# The Use of Neural Network Modeling Methods to Determine Regional Threshold Values of Hydrochemical Indicators in the Environmental Monitoring System of Waterbodies

**DOI:** 10.3390/s23136160

**Published:** 2023-07-05

**Authors:** Yulia Tunakova, Svetlana Novikova, Vsevolod Valiev, Evgenia Baibakova, Ksenia Novikova

**Affiliations:** 1Department of General Chemistry and Ecology, Kazan National Research Technical University Named after A.N. Tupolev-KAI, 10 K. Marx St., Kazan 420111, Russia; 2Department of Applied Mathematics and Computer Science, Kazan National Research Technical University Named after A.N. Tupolev-KAI, 10 K. Marx St., Kazan 420111, Russia; svnovikova@kai.ru; 3Research Institute for Problems of Ecology and Mineral Wealth Use of Tatarstan Academy of Sciences, 28 Daurskaya St., Kazan 420087, Russia; ipen-anrt@mail.ru

**Keywords:** surface water, environmental monitoring, hydrochemical indicators, neural network technologies, regional research project standards

## Abstract

The regulation of the anthropogenic load on waterbodies is carried out based on water quality standards that are determined using the threshold values of hydrochemical indicators. These applied standards should be defined both geographically and differentially, taking into account the regional specifics of the formation of surface water compositions. However, there is currently no unified approach to defining these regional standards. It is, therefore. appropriate to develop regional water quality standards utilizing modern technologies for the mathematical purpose of methods analysis using both experimental data sources and information system technologies. As suggested by the use of sets of chemical analysis and neural network cluster analysis, both methods of analysis and an expert assessment could identify surface water types as well as define the official regional threshold values of hydrochemical system indicators, to improve the adequacy of assessments and ensure the mathematical justification of developed standards. The process for testing the proposed approach was carried out, using the surface water resource objects in the territory of the Republic of Tatarstan as our example, in addition to using the results of long-term systematic measurements of informative hydrochemical indicators. In the first stage, typing was performed on surface waters using the neural network clustering method. Clustering was performed based on sets of determined hydrochemical parameters in Kohonen’s self-organizing neural network. To assess the uniformity of data, groups in each of the selected clusters were represented by specialists in this subject area’s region. To determine the regional threshold values of hydrochemical indicators, statistical data for the corresponding clusters were calculated, and the ranges of these values were used. The results of testing this proposed approach allowed us to recommend it for identifying surface water types, as well as to define the threshold values of hydrochemical indicators in the territory of any region with different surface water compositions.

## 1. Introduction

The environmental monitoring of surface waterbodies in urbanized territories is carried out to obtain, assess, and forecast the variability of hydrochemical indicators and provide information and analytical support for the regulation of the anthropogenic load on surface waters. It is known that the component composition of natural waters can be determined by both the natural hydrochemical background and anthropogenic load. The regulation of the anthropogenic load is carried out by establishing the standards of surface water quality, as well as the standards of permissible exposure; for the Republic of Tatarstan, both of these standards are regulated in [1]. Currently, the applied standards have been approved in the territory of every state. For example, in the territory of the Republic of Tatarstan, the maximum permissible concentrations (MPC) that are used were developed for the entire territory of the Russian Federation without taking into account regional specifics. 

However, with the development of basin approaches for the assessment and management of water resources, as well as both the accumulation of actual data on the differences in hydrochemical indicators in the local area’s waterbodies and the improvement of analytical capabilities for their measurement, it has become necessary to implement fundamentally new approaches to normalize surface water quality at a regional level [2,3,4].

However, at present, there are no uniform methodological approaches to developing such standards in either the practice of hydrochemical assessments or the protection of water bodies. The problem is that there is no methodology to determine them for surface waterbodies that can reasonably answer the key questions that arise in the practice of developing regional standards, namely, identifying which principle can be used as the basis for determining the threshold values of hydrochemical indicators, while also reflecting the conditions for the formation of local hydrochemical characteristics since the value of the regional standard largely depends on this. It is possible to single out the main recommendations for the development of such standards, which are reflected in [2,3,4,5,6,7] in particular as follows: −Standards should be determined only based on experimental studies in real conditions;−Special attention should be paid to the selection of the most informative indicators, characterizing the condition of the investigated waterbody first of all;−Regional regulations should exceed neither national nor WHO (World Health Organization) standards.

Currently, when calculating the permissible exposure standards for the territory of the Russian Federation, only the hydrological criteria for the multiplicity of dilution and the determination of background (confidence) concentrations of pollutants in specific areas of the waterbody are assessed [8,9]. The current system of calculating quality standards and impact does not take into account other natural features.

The regional threshold values of the indicators concerning a specific river basin should be developed in such a way that water protection measures for surface waterbodies belonging to the same region can be coordinated and environmentally sound, which is justified in [10,11,12], for example.

It should be noted that in several countries, a basin approach has been used in the development of water quality standards. In particular, the standards of a maximum permissible anthropogenic load on waterbodies used in the USA (Total Maximum Daily Load—TDML) [13,14,15] are being developed with the involvement of geoinformation technologies, implementing a basin approach when developing standards for permissible impacts, and are based on the results of both long-term monitoring studies and mathematical modeling. To take into account the natural features of waterbodies when rationing, it is necessary to take into account the experience of the EU countries and North America (Canada, USA, Mexico), which can be summarized as follows:-Territories are divided into sections with relatively homogeneous natural conditions;-The waterbodies within these sections are classified according to their key features;-For each element of the classification, the standard of the waterbody is determined, which can then be labeled as the regional quality standard [12,13,14]. International experience in taking into account natural features makes it possible to build a statistical classification of surface water rationing within river basins of the first order, using an assessment of the spatial and seasonal variability of their hydrochemical conditions. Along with the recognition of the need to use regional standards, it is important to note that the value of the permissible load cannot remain constant from year to year but depends on the hydrological regime of watercourses and the conditions of the formation of the natural hydrochemical background. Regional standards of quality and impact should be considered as a dynamic value; therefore, there is always a possibility of exceeding normative water quality [16,17,18]. This means that the establishment of regional threshold values of hydrochemical indicators can be determined by a given level of security for maintaining the required water quality or the probability of exceeding it, which means that the normative value lies in a certain range, the scope of which is set by specific regional characteristics.

Some authors suggest using, in particular, a 30% increment to the background concentration, with a range of Sf–1.3 Sf, to determine the magnitude of the range of values for regional threshold concentrations [3,19]. In our opinion, any approaches involving certain empirical coefficients are speculative, difficult to justify, and unnecessary since all the necessary estimates can be carried out statistically on a sufficiently large factual material, using a quartile scale or variance for this.

The problem of identifying ranges of regional quality standards that take into account the variability of the state of a waterbody can be solved by fixing relatively stable time and hydrochemical space indicators using cluster analysis. Cluster analysis allows you to decompose data with the property of “similarity” in a given sense and separate them from “dissimilar” data. The decomposition of data should have the property of homogeneity in groups, that is, data within the same group should be as similar as possible to each other; it should also have heterogeneity between groups, that is, data belonging to different groups should differ from each other as much as possible [20,21]. The results of the analysis of samples on hydrochemical indicators of natural waters, when allocated to different clusters and reflecting different modes of functioning in a waterbody, should be assigned their regional threshold values of hydrochemical indicators.

Therefore, in [22], the authors applied the McKean K-means clustering algorithm to identify the substances polluting seawater. The paper showed the possibility of a clear separation of water samples by contamination groups with metals and phenols from uncontaminated water and the identification of the contaminant based on the proposed clustering. Other scientists have also obtained positive results from applying the k-means clustering algorithm. In particular, in [23], the authors successfully used this algorithm to determine water pollution zones in Indonesia. The disadvantage of this method was the need to know the number of clusters into which the data would be subsequently divided.

The so-called hierarchical clustering methods, for example, Ward’s hierarchical cluster analysis method, are devoid of such a disadvantage [24]. This procedure is also successfully used in environmental monitoring and management tasks to assess the risk of contamination of individual soil areas [25], evaluate the quality of water in rivers [26], study dust and air pollution [27], etc. The authors [28] applied hierarchical clustering to the group time series of the results with the environmental monitoring of surface waters in one particular river, showing that cluster analysis was applicable for grouping data not only to identify homogeneous spatial areas of pollution but also temporarily.

Machine learning-based techniques are being increasingly used to process environmental monitoring results, for example, Kohonen neural networks for grouping data [29]. The main advantage of neural network technologies is their ability to learn and retrain. This allows you to flexibly adjust the model without having to completely rebuild it, unlike, for example, hierarchical clustering. It is enough to retrain the neural network model with new data for several epochs. Since the model has already been trained, remembering the new data is fast. In this case, the structure of the network does not change, but the weight coefficients adjust, and the neural network model works in a new way. Thus, Kohonen networks were used in [30] to model the pollution level of the air basin of cities. In [31], the authors used Kohonen networks to identify the main patterns of soil pollution. In [32], the authors modeled the areas of groundwater pollution.

Neural network technologies are widely used for other tasks in the environmental monitoring of water bodies. Thus, the dependencies of the content of pollutants in surface waterbodies on the parameters of discharge sources, hydrochemical conditions, and other influencing factors have an approximate analytical expression and are poorly studied in nonlinear dependencies. In this regard, modeling such dependencies based on intelligent learning neural network structures has excellent prospects. Direct propagation networks, or perceptrons, are mainly used for such tasks. Thus, in [33,34,35], the parameters of the state of river surface waters were modeled in [36]—groundwater. Other paradigms of neural networks were used much less often. Thus, in [37], recursive LSTM networks were used to simulate the biological contamination of reservoirs by algae. In [38], the authors used Hopfield’s self-associative networks to assess the sustainability of the urban ecosystem. Attempts to use CNN deep convolutional neural networks, which are popular today, are still limited to recognizing particular ecological and biological objects in the natural environment [39,40,41,42,43].

Thus, we set this study’s goal—to develop an approach for determining regional standards for surface water quality using intelligent neural network models based on a series of experimental observation data to take into account natural features and the spatiotemporal variability of the hydrochemical state of water bodies.

## 2. General Description of the Idea of a Method for Determining Regional Threshold Values

We propose a fundamentally new approach to developing regional quality standards based on the threshold values of informative hydrochemical indicators using statistical research, expert evaluation, and an empirical approach.

In the first stage, the available sets of measured hydrochemical indicators in various waterbodies of a particular water basin were clustered. That is, they were divided into relatively homogeneous groups. Experts in the subject area assessed the degree of detail in the partition. As a result, water types were determined with homogeneous, according to experts, hydrochemical indicators and with a necessary degree of detail.

Kohonen neural networks were chosen as a clustering algorithm and designed to isolate homogeneous data groups with any degree of accuracy based on an algorithm similar to the k-means algorithm. Like any neural network model, the Kohonen network could retrain new data without being rebuilt. In addition, the advantage of this model was the simplicity of visualization, which simplifies qualitative interpretation for experts in the future [44]. As a result of reconnaissance experiments comparing the accuracy of clustering algorithms (K-means, hierarchical agglomerative and divisive clustering, and SOM), the accuracy of the SOM algorithm turned out to be 11% higher. This comparison was carried out according to the criteria of separability (FM-criterion) and density (SWC-criterion) of clustering [45].

The proposed method of neural network clustering for allocating regional threshold values of hydrochemical indicators consists of three computational blocks and two blocks for checking conditions. The first computational block is a single clustering algorithm based on Kohonen neural networks, followed by two-dimensional visualization with SOM maps, which performs the preprocessing of input vectors. Preprocessing consists of a rough division of the initial data into groups. This algorithm can be used independently and as an integral element of the second computing unit.

The second computing block is a tree of sequential clustering algorithms based on Kohonen networks that can perform the in-depth detailing of groupings as needed and are used for more accurate batches of data.

The third computing unit is auxiliary, which encodes data of the “date-time” type for the adequate calculation of distances between clustered vectors, as a rule, considering seasonality. The complexity of calculations with such data lies in the impossibility of applying standard arithmetic operations to them directly. For example, when analyzing the distances between three dates: 5 January, 15 March, and 23 December, direct calculations of the difference show that the dates of 5 January and 15 March are much “closer” to each other than the dates of 5 January and 23 December, although, from the point of view of the time of year, both dates refer to the concept of “winter” and are separated from each other by 12 days. To adequately calculate the distances between such data vectors, a unique coding method has been developed to represent each date as a four-dimensional vector. Each element of such a vector indicates to what extent the stated date belongs to one of the four seasons. To adequately calculate the distances between such data vectors, a unique coding method was developed to represent each date as a four-dimensional vector. Each element of such a vector indicates to what extent the stated date belongs to one of the four seasons.

Condition-checking blocks were used to switch between computing blocks. In particular, the first conditional block determined whether pre-coding was required, that is, whether an auxiliary computing block could be used; the second conditional block determined the need for direct or sequential clustering, that is, switching between the blocks of the first and second types.

This method thus implemented a step-by-step grouping of data depending on the specified conditions. Graphically it could be represented in the form (Figure 1).

### 2.1. The Block of Direct Neural Network Clustering

It is a two-layer self-organizing neural network of the “Kohonen network” type considering the way neurons are arranged in a two–dimensional rectangular grid. It is trained according to a modified winner-takes-all algorithm with a consistently decreasing neighborhood measure according to the formula:(1)$$\Delta w_{i^{*}}^{T}=\eta \Lambda (|i-i^{*}|)\left(x^{T}-w_{i^{*}}\right)$$

Here, $\Delta w_{i^{*}}^{T}$ is the increment of the weight vector of neuron *i**; *i** is the winning neuron for the input vector *x*; $w_{i^{*}}$ is the weight vector of neuron *i**. The winning vector is selected from the condition that its vector of weights $w_{i^{*}}$ is closer to this input vector *x* than all other neurons: $\left|w_{i^{*}}-x\right|\leq \left|w_{i}-x\right|$ for all *i*. $\left|i-i^{*}\right|$ is the distance from an arbitrary neuron I to the winning neuron.

$\Lambda (|i-i^{*}|)$ is the neighborhood function. This function is equal to one for the winning neuron with index *i* * and decreases with distance according to the law:(2)$$\Lambda (a)=\exp\left(\frac{-a^{2}}{\sigma^{2}}\right)$$

*η*—learning rate, *σ* is the radius of the interaction of neurons. 

### 2.2. The Block of Phased Neural Network Clustering

This block builds a tree of interconnected neural network clustering algorithms with an increase in the degree of grouping accuracy at each level of the tree. The construction is based on the repeated repetition of the clustering procedure by the Kohonen network for data groups with insufficient granularity. In the first stage, all data a priori belong to a single cluster (group). According to the expert, if data in the cluster are not a homogeneous group, the group is divided into a given number of clusters using the Kohonen neural network. Thus, clusters of the first level of the tree are formed. Next, the expert analyzes the data in each selected cluster (group). If the analyst also finds pronounced heterogeneity for one of the groups, these groups, in turn, are also divided into subclusters by the Kohonen network. As a result, clusters of the second level of the tree are formed, the detail of which is more significant than in the first level. The analysis of groups obtained as a result of clustering and their further clustering (formation of cascade levels with an increase in the degree of detail) continued until an acceptable degree of uniformity was achieved for all the selected groups, according to the expert.

An expert could be either a person—a specialist in the subject area, or an automaton or algorithm that evaluates the degree of homogeneity for data in a group through mathematical or statistical analysis methods.

### 2.3. Pseudo-Fuzzy Coding Block

The pseudo-fuzzy encoding of data for the “date-time” type is an algorithm that replaces a single date-time value with a vector of four values, each expressing a degree of belonging to a specified date and to one of the four seasons. As a result of encoding, date x is represented by a vector (sequence) *Y* = (*y*_1_, *y*_2_, *y*_3_, *y*_4_), where 1, 2, 3, 4 are the codes of the seasons, $y_{i}=f^{{}_{i}}(x)$ is the degree of belonging that *x* has to the season *i*, and $f^{{}_{i}}(x)$ is the coding function for the season *i*:(3)$$f^{{}_{i}}(x)=\exp\left[-{\left(\frac{x-m_{i}}{\sigma_{i}}\right)}^{2}\right]$$

*m_i_* is a numeric value of the “date-time” type that mostly corresponds to the season *i*; $\sigma_{i}=\frac{b_{i}}{3}$; and *b_i_* is the maximum deviation of the date from the value *m_i_*, at which the date still corresponds to the season *i*. 

## 3. Conducting Computational Experiments

For testing the proposed algorithm, the values of hydrochemical quality indicators for the surface waterbodies were used at 12 sampling points in the period from March 2014 to December 2021: Kuibyshev reservoir, 4.7 km below Kazan City;Volga River, above Zelenodolsk City;Ashit River, Alan-Bexer village;Volga River, Kazan city, 1 km above the water intake;Volga River, KzylBayrak village;Kazanka River, 3rd transport dam;Kazanka River, Usady village;Kama River, Sorochy Gory village;Mesha River, Karaduli village;Mesha River, Uzyak village;Sviyaga River, the bridge on the M 7 highway;Sulitsa River Savino village.

We analyzed 26 names of informative hydrochemical indicators characterizing the quality of natural waters: aluminum, mg/L; ammonium ion, mg/L; BOC5 (Biochemical oxygen consumption), mgO2/L; Suspended solids, mg/L; Bicarbonates, mg/L; Iron, mg/L; Hardness, degree of hardness; Calcium, mg/L; Oxygen solution, mgO2/L; Magnesium, mg/L; Manganese, mg/L; Copper, mg/L; Petroleum products, mg/L; Nickel, mg/L; Nitrates, mg/L; Nitrites, mg/L; Total ion content (mineralization), mg/L; Transparency, cm; Sulfates, mg/L; Temperature, C; Phenol, mg/L; Phosphate ion, mg/L; COD (chemical oxygen consumption), mg/L; Chlorides, mg/L; Zinc, mg/L; Electrical conductivity, mcm/sm. We used the IR spectrometry method to determine oil products; the content of metal ions—atomic absorption spectrometry; phenol—gas-liquid chromatography; chloride, sulfate, and nitrate ions—ion chromatography. We also used appropriate certified analysis methods to determine the values of the remaining indicators. We performed calculations using special software for data analysis. The unique identifier of each data set was the coordinate of the sampling point. An additional factor taken into account in the model was seasonality.

### 3.1. The Result of Grouping Based on Direct Neural Network Clustering without Taking into Account Seasonality

The data were grouped into four clusters (Figure 2). Here, “0”, “1”, “2”, and “3” were the number of calculated clusters.

The main distinguishing features of the selected groups included the content of calcium, magnesium, sulfates, and electrical conductivity (Figure 3).

The distribution of data across clusters was relatively uniform. Based on the data obtained, it could be concluded that lower values of the controlled indicators at sampling points formed clusters 0 and 1. By contrast, the water quality of cluster 0 was slightly higher than 1. Cluster 2 occupied an intermediate position, while the waters of Cluster 3 had the highest values of the controlled indicators.

### 3.2. The Result of Grouping Is Based on Phased Neural Network Clustering Taking into Account Seasonality

Next, hydrochemical indicators were grouped, considering seasonality and using the Kohonen neural self-learning network. In the first stage, four clusters were identified (Figure 4). Here, “0”, “1”, “2”, and “3” were the number of calculated clusters.

A team of experts in this subject area was formed, including employees of the Institute of Ecology and Subsoil Use Problems of the Academy of Sciences of the Republic of Tatarstan, who conducted long-term monitoring studies on the studied waterbodies and determined the background values of the hydrochemical parameters. An analysis of the visualized results of the distribution of hydrochemical indicators by clusters depending on the seasons made it possible to conduct an expert analysis. The water test dates were coded in a pseudo-fuzzy encoding block. The strong influence of the seasons on data distribution across clusters can be seen (Figure 5). Here, the blue color symbolized the weak belonging of the data to the specified season, and the red color symbolized a very strong belonging. The remaining colors symbolized intermediate positions from weak to strong—from blue to red.

Based on expert analysis at this stage, a conclusion was made about the random and uniform nature of the content in the surface waters of such components as ammonium ion, suspended solids, Iron, Nickel (single points of contamination), Phenol (single points of contamination), and Zinc (single points of contamination). For example, Figure 6 shows the data regarding the ammonium distribution by clusters. It could be seen that the ammonium content in all clusters was approximately the same. A similar distribution type of concentration by clusters is also characteristic of other listed substances. These indicators were excluded from further analysis.

According to expert analysts, the data sets in cluster No. 1 needed to have a greater degree of uniformity. Therefore, it was decided to split this cluster into subclusters. The number of subclusters was determined based on the visually selected heterogeneous zones of the cluster after visualization (Figure 7).

Cluster 1 was split into three subclusters (Figure 8). 

The further analysis of the visualization of the grouping by SOM-maps allowed us to draw an unambiguous conclusion about the parameters characterizing the mineralization of water (sulfates, bicarbonates, hardness, calcium, and electrical conductivity) as the main criteria determining hydrochemical variability (Figure 9). 

The separation of the values of iron, petroleum products, phosphorus, manganese, suspended solids, and oxygen was less pronounced. There were no clear distribution patterns across clusters for the rest of the indicators.

Here, cluster numbers “1-0” mean “subcluster zero of the first cluster”, cluster numbers “1-1” mean “first subcluster of the first cluster”, and finally, cluster numbers “1-2” mean “second subcluster of the first cluster”.

As a result, we built the final model of a tree with six leaves: Cluster 0, Cluster 1-0, Cluster 1-1, Cluster 1-2, Cluster 2, and Cluster 3 (Figure 10). Accordingly, experts were asked to identify six qualitative assessments—the types of surface waters covering the entire variability of the background hydrochemical composition of the water in the Volga-Kama basin.

Descriptive statistics of variational data series made identifying the ranges of values characterizing a particular cluster of indicators possible. The statistical significance of these differences was assessed using the nonparametric Kraskel–Wallis criterion, followed by a paired Mann–Whitney evaluation. The intercluster values of the Kraskel–Wallis criterion for the leading indicators (typifying one or another type of water in the description) were pretty significant (*p* = 0.056–0.001). When describing specific clusters, the threshold values of indicators that characterized different types of waters in one way or another were given. In contrast, the upper threshold corresponded to the upper quartile (75%) of the variation series and was accepted as a regional threshold value that was defined for each type of water.

## 4. Discussion 

For each cluster list, corresponding samples of hydrochemical indicators were formed, and their statistical processing was carried out. As a result of the cluster analysis and expert evaluation, the types of waters that differed in their values of hydrochemical indicators were identified:

Water type 1 (Cluster 0). Calcium-magnesium bicarbonate water of high mineralization (no more than 1095 mg/L), high hardness (no more than 13.7 mg-eq./L), with a high oxygen content (no less than 8.5 mgO_2_/L).

Water Type 2 (Cluster 1). Bicarbonate calcium-magnesium water of moderate mineralization (no more than 276.5 mg/L), low hardness (no more than 3.3 mg-eq./L), with an average oxygen content (no less than 7.41 mgO_2_/L).

Water Type 3 (Cluster 2). Bicarbonate calcium-magnesium water of medium mineralization (no more than 640 mg/L), increased hardness (no more than 7.93 mg-eq./L), with a high oxygen content (no less than 8.2 mgO_2_/L).

Water Type 4 (Cluster 3). Bicarbonate calcium-magnesium water of high hardness (not less than 13.1 mg-eq./L), high mineralization (not more than 960 mg/L), with an average oxygen content (not less than 7.9 mgO_2_/L).

Water type 5 (Cluster 4). Bicarbonate calcium-magnesium water of medium hardness (no more than 11.3 mg-eq./L), moderate mineralization (no more than 850 mg/L), with a high oxygen content (no less than 9.7 mgO_2_/L).

Water type 6 (Cluster 5). Bicarbonate calcium-magnesium water of medium hardness (no more than 8.38 mg-eq./L), moderate mineralization (no more than 670 mg/L), with a low oxygen content (no more than 6.9 mgO_2_/L).

The threshold values of the remaining estimated hydrochemical indicators for the selected types of water are given in Table 1.

## 5. Conclusions

The quality of surface waters, even within sections of one water basin, can be characterized by significant spatial heterogeneity due to the regional peculiarities of the composition of natural waters. The analysis of the values of the obtained hydrochemical indicators for the sections of the Volga-Kama basin in the territory of the Republic of Tatarstan confirmed this. Therefore, to regulate the anthropogenic load and determine the impact standards on waterbodies, it is necessary to develop an approach that can determine the regional threshold values of measured hydrochemical indicators. Regional quality standards for substances found in natural waters in relatively high or low concentrations should be developed to preserve the water composition of waterbodies formed under the influence of natural factors. The results of measurements of hydrochemical indicators and the use of a unique pseudo-fuzzy coding technique could make it possible to consider the distribution of hydrochemical indicator measurements adequately. To determine the characteristic features of each group (cluster), we must subject the values of hydrochemical parameters in the group to expert statistical analysis. Then, based on this analysis, it would become possible to identify the most significant indicators, determine regional threshold contents, carry out target rationing, and obtain critical assessments of the state of the water body. Since there is no automated spatial analysis of the distribution of monitoring data in the current environmental monitoring system of surface water bodies, an expert assessment is required. The spatial and physical-geographical differentiation of different types of waters is set by experts at the stage of detailing the cluster analysis. In the future, in the indicated scheme of neural network modeling, the external expert assessment should be entirely replaced by the corresponding system algorithm. 

Thus, the presented typification generalizes the hydrochemical features of local sites, and the fact that their spatial and physical-geographical location conditions often turn out to be similar for one type of water only confirms the effectiveness of the proposed approach. Regional threshold values are characteristics of the various experimentally measured hydrochemical parameters. Therefore, as a rule, they do not exceed the accepted MPC and are flexible regulatory values. It should also be noted that for several substances, for example, hydrocarbonates, there are no standards. This lack of standards is because such substances are prominent representatives of substances of natural origin, and their content in natural waters varies greatly depending on the geochemical features of a specific basin water area. The proposed approach, as well as regional regulations in general, aim to solve the problem of accounting for the concentrations of substances that are of natural origin.

The approach we proposed applies to the typification of other natural objects and the determination of threshold values of measured indicators with the development of quality and impact standards and considering nature features.

## Figures and Tables

**Figure 1 sensors-23-06160-f001:**
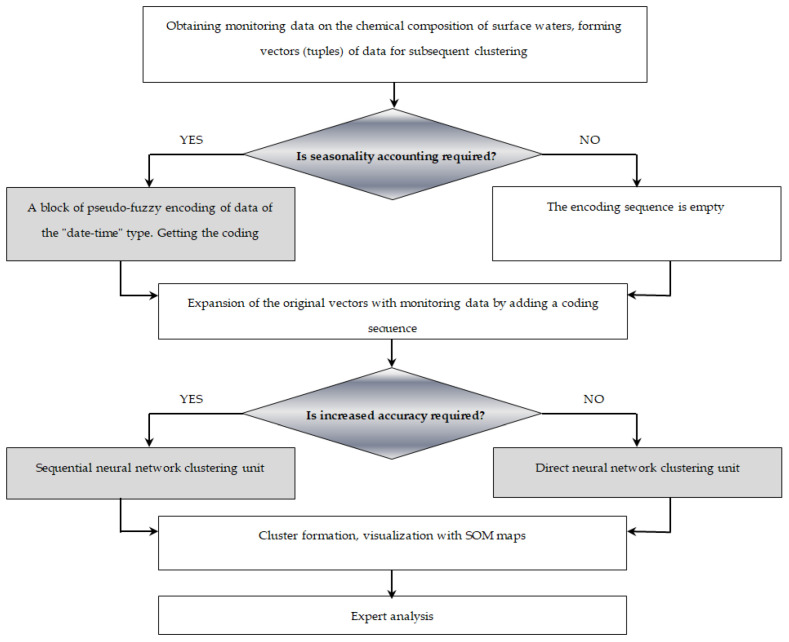
Structure of interaction of blocks for the method of conditional step-by-step grouping of monitoring data.

**Figure 2 sensors-23-06160-f002:**
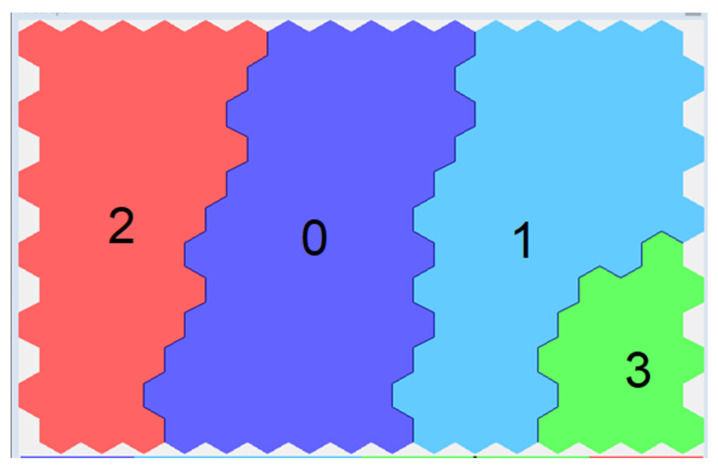
Clusters arrangement on the SOM-map.

**Figure 3 sensors-23-06160-f003:**
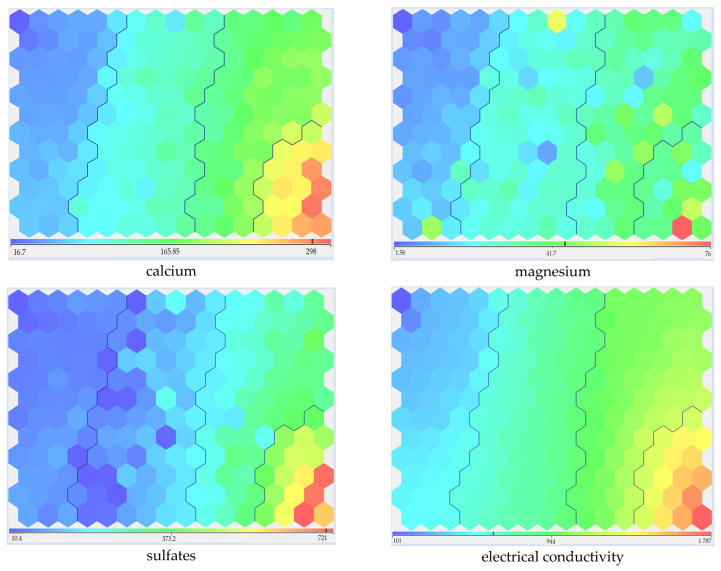
Distribution of main indicators by clusters.

**Figure 4 sensors-23-06160-f004:**
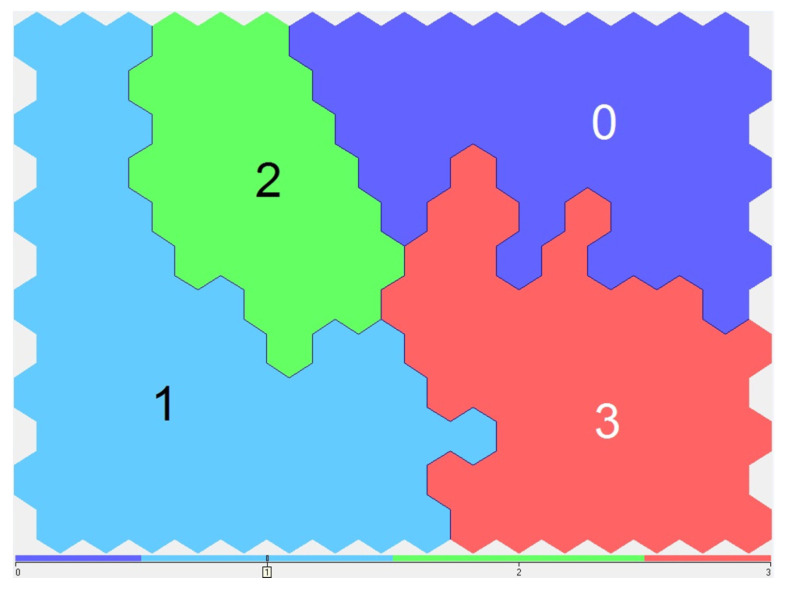
Clusters arrangement on the SOM-map seasonally adjusted.

**Figure 5 sensors-23-06160-f005:**
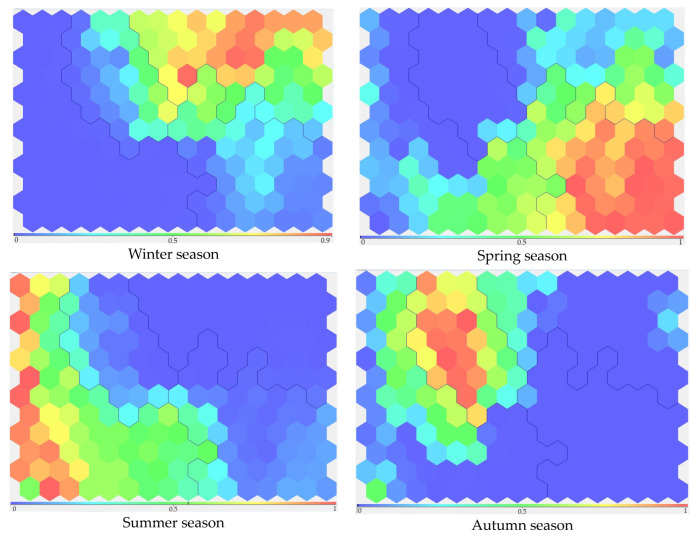
Distribution of hydrochemical indicators by clusters depending on the seasons.

**Figure 6 sensors-23-06160-f006:**
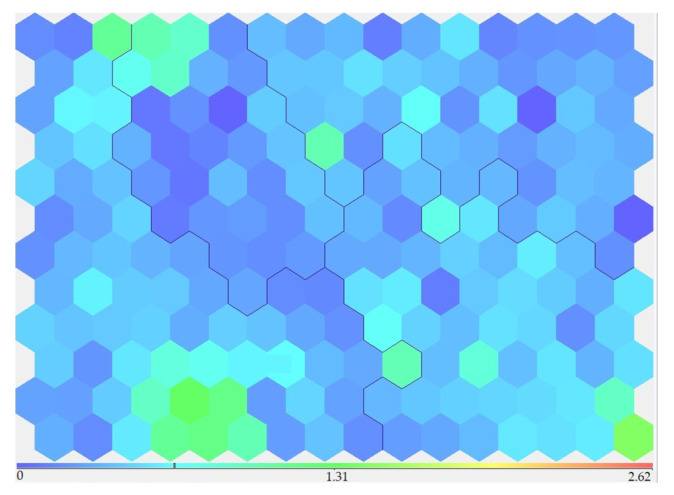
Distribution of ammonium ion by clusters.

**Figure 7 sensors-23-06160-f007:**
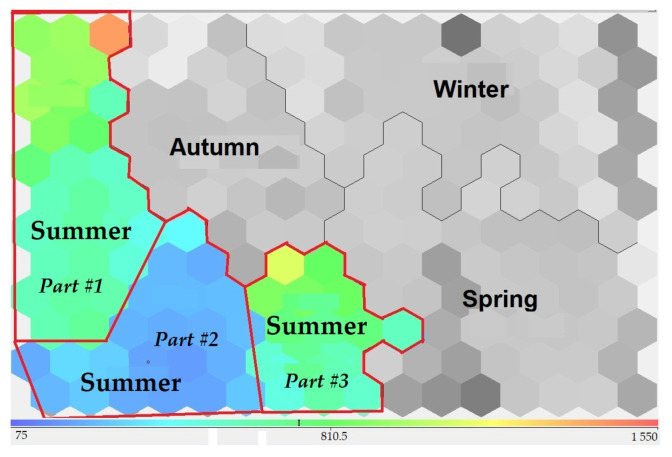
Three visually distinct parts to cluster 1 (summer). Mineralization parameter.

**Figure 8 sensors-23-06160-f008:**
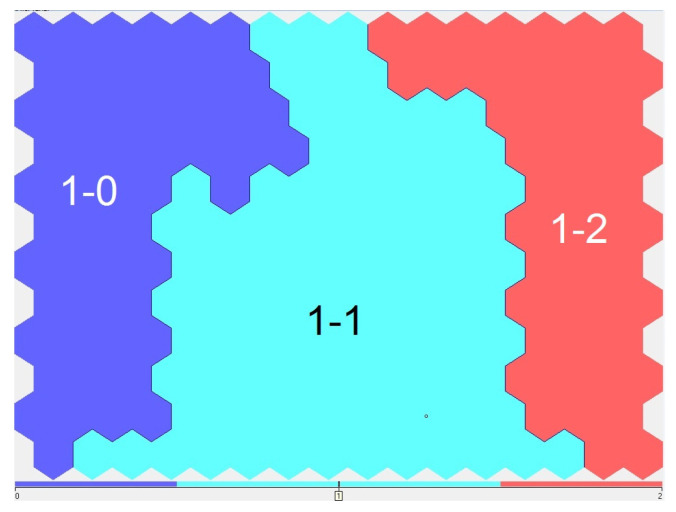
Three sub-clusters of cluster 1 “Summer”.

**Figure 9 sensors-23-06160-f009:**
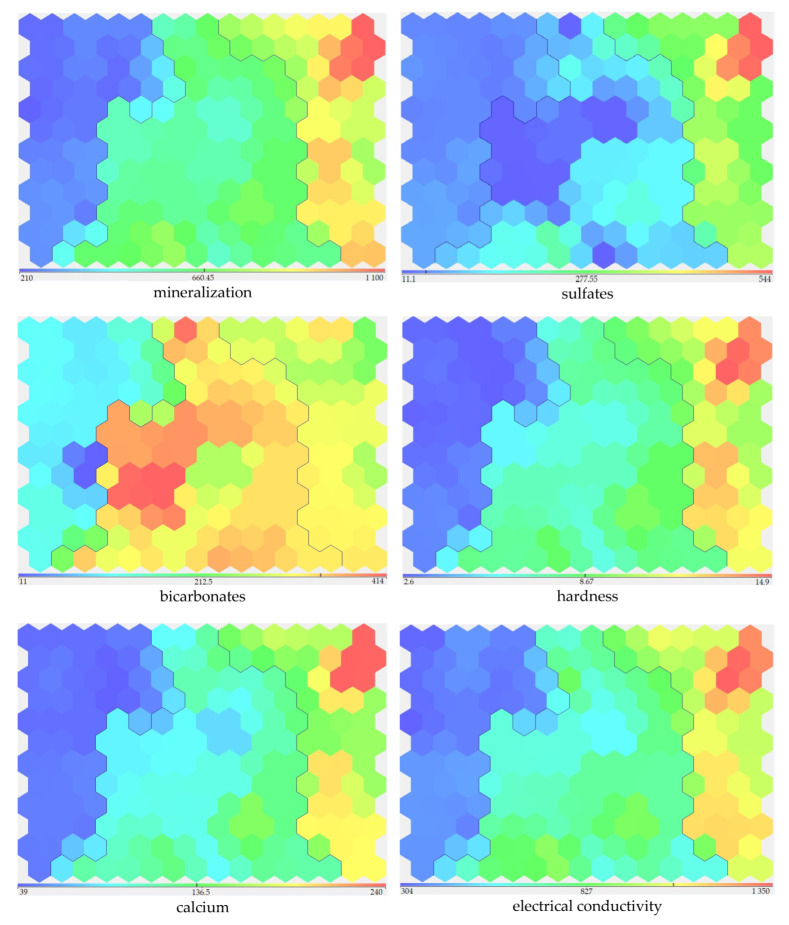
Distribution of indicators by subclusters.

**Figure 10 sensors-23-06160-f010:**
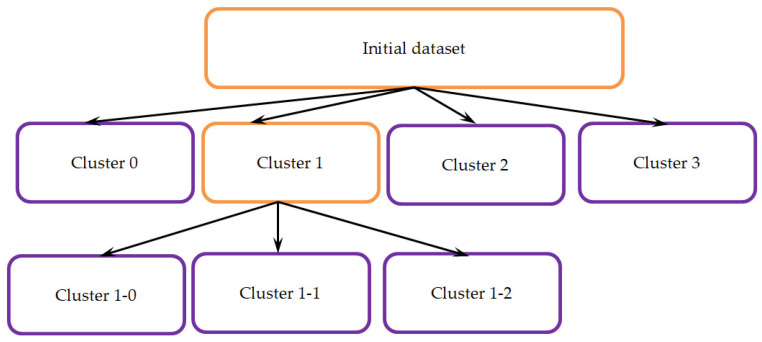
The structure of the sequential neural network grouping tree.

**Table 1 sensors-23-06160-t001:** Threshold values of hydrochemical indicators for selected types of natural water.

Type of Natural Waters	Threshold Values of Hydrochemical Parameters, mg/L								
	HCO_3_^−^	Ca^2+^	Mg^2+^	Na^+^ + K^+^	Fe^2+^	Petroleum Products	SO_4_^2−^	PO_4_^3−^	Cl^−^
1	365	223	44.0	58	0.082	0.078	391	0.247	21.3
2	127	48	10.6	16	0.139	0.025	66.4	0.259	26.1
3	365	117	26.6	25	0.077	0.050	141	0.420	15.7
4	315	199	35.2	30	0.094	0.040	371	0.390	18.5
5	342	164	36.6	25	0.079	0.040	301	0.284	18.0
6	300	117	23.6	25	0.143	0.047	173	0.428	19.7

## Data Availability

The data presented in this study are available on request from the corresponding author. The data are not publicly available due to the rules of our contract conditions with our customers.

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
