# Peer review of "The Use of Neural Network Modeling Methods to Determine Regional Threshold Values of Hydrochemical Indicators in the Environmental Monitoring System of Waterbodies"

_sensors, 2023, doi:10.3390/s23136160_

Round 1

Reviewer 1 Report

Dear All,

The manuscript could be accepted after minor revision.

Best Wishes

Author Response

Dear Reviewer! We have made corrections to the abstract, the introduction, and corrected remarks on the design.

 In order to take into account the natural features of water bodies when rationing, it is necessary to take into account the experience of the EU countries and North America (Canada, USA, Mexico):

- divide the territory into sections with relatively homogeneous natural conditions;

- classify water bodies according to key features;

- for each element of the classification to determine the reference water bodies;

- based on observational data on reference water bodies, establish regional quality standards.

Thus, we have set the goal of the study to develop an approach for determining regional standards for surface water quality using intelligent neural network models to take into account natural features and variability of the state of water bodies.

Regional threshold values represent characteristics of variation series of actually measured hydrochemical parameters therefore, as a rule, do not exceed the accepted MAC and are though more strict, but flexible regulating values. At the same time, it should be noted that in some cases, for example, when assessing the content of substances of natural and anthropogenic origin, such as iron, manganese, and sulfate, such thresholds in some river basins may exceed MPC due to the specifics of formation of the natural local hydrochemical regime.

Expert evaluation is a necessary element of the proposed system first of all due to the absence in it of automated spatial analysis of data distribution. The problem of spatial analysis will be solved during further development of the approach, in particular, by introducing the algorithm of estimation of geometrical distances between sampling points of estimated indicators and considering this variable during the clustering of different water bodies. Therefore, in the present study, homogeneity of clustering taking into account the spatial distribution of sampling points was evaluated by experts. In other words, the expert considers the quality of clustering by evaluating the clustering by clusters of the spatial localization of sampling locations represented by the data. In cases where the data in the cluster does not represent geographically contiguous localization, a decision is made about the lack of detail. Unfortunately, an effective IT solution for neural network generalization of such information as part of the developed system at this stage of research is not provided but is actively being developed. The current version of the system implements only automatic neural network estimation of temporal heterogeneity.

The second important function of expert estimation is the estimation of statistical homogeneity of data grouped into selected clusters. Such statistical evaluation was performed by determining the statistical significance of inter-cluster differences between the centers of distributions of each indicator. This stage has not yet been implemented in automatic mode and requires the participation of a specialist.

Due to the fact that these procedures are rather routine sequences of actions, a detailed description of the results of each step would significantly increase the volume and complicate the perception of the presented material, we have limited ourselves to a sufficient listing of necessary computational procedures, such as the Kruskal-Wallace and Mann-Whitney tests, whose separate description is beyond the essence and purpose of the study.

 In the future, in the mentioned scheme of neural network modeling, external expert assessment will be completely replaced by an appropriate system algorithm.

Reviewer 2 Report

This paper should be accepted, but authors must make major revisions and improvements, mainly related to introduction and significantly improve the point 2 and conclusions, which will make their work more valuable. Also, the must significantly revise the formatting of bibliographic references and a better synthesis of ideas. I suggested corrections (pdf file) which should allow a better understanding and improve on the style.

Minor editing of English language required

Author Response

Dear Reviewer! We have made corrections to the abstract, the introduction, and corrected remarks on the design.

Thus, we have set the goal of the study to develop an approach for determining regional standards for surface water quality using intelligent neural network models to take into account natural features and variability of the state of water bodies.

The introduction section has been finalized in view of the proposed publications

 In order to take into account the natural features of water bodies when rationing, it is necessary to take into account the experience of the EU countries and North America (Canada, USA, Mexico):

- divide the territory into sections with relatively homogeneous natural conditions;

- classify water bodies according to key features;

- for each element of the classification to determine the reference water bodies;

- based on observational data on reference water bodies, establish regional quality standards.

Regional threshold values represent characteristics of variation series of actually measured hydrochemical parameters therefore, as a rule, do not exceed the accepted MAC and are though more strict, but flexible regulating values. At the same time, it should be noted that in some cases, for example, when assessing the content of substances of natural and anthropogenic origin, such as iron, manganese, and sulfate, such thresholds in some river basins may exceed MPC due to the specifics of formation of the natural local hydrochemical regime.

Expert evaluation is a necessary element of the proposed system first of all due to the absence in it of automated spatial analysis of data distribution. The problem of spatial analysis will be solved during further development of the approach, in particular, by introducing the algorithm of estimation of geometrical distances between sampling points of estimated indicators and considering this variable during the clustering of different water bodies. Therefore, in the present study, homogeneity of clustering taking into account the spatial distribution of sampling points was evaluated by experts. In other words, the expert considers the quality of clustering by evaluating the clustering by clusters of the spatial localization of sampling locations represented by the data. In cases where the data in the cluster does not represent geographically contiguous localization, a decision is made about the lack of detail. Unfortunately, an effective IT solution for neural network generalization of such information as part of the developed system at this stage of research is not provided but is actively being developed. The current version of the system implements only automatic neural network estimation of temporal heterogeneity.

The second important function of expert estimation is the estimation of statistical homogeneity of data grouped into selected clusters. Such statistical evaluation was performed by determining the statistical significance of inter-cluster differences between the centers of distributions of each indicator. This stage has not yet been implemented in automatic mode and requires the participation of a specialist.

Due to the fact that these procedures are rather routine sequences of actions, a detailed description of the results of each step would significantly increase the volume and complicate the perception of the presented material, we have limited ourselves to a sufficient listing of necessary computational procedures, such as the Kruskal-Wallace and Mann-Whitney tests, whose separate description is beyond the essence and purpose of the study.

 In the future, in the mentioned scheme of neural network modeling, external expert assessment will be completely replaced by an appropriate system algorithm.

Reviewer 3 Report

The topic is interesting and developing. However, the scientific content is poor and shallow. Therefore, the manuscript requires deep corrections and must be deeply reworked with a clear and scientific approach. In addition, the manuscript, as it stands, requires several deep revisions because it had flaws. The topic is interesting and developing. However, the scientific content is poor and shallow. Therefore, the manuscript requires deep corrections and must be deeply reworked with a clear and scientific approach. In addition, the manuscript, as it stands, requires several deep revisions because it had flaws. 

I proofread the manuscript with some notes that could be useful to the authors to revise, update, and improve a strengthened and balanced version of the manuscript. (see attached file)

Author Response

Dear Reviewer!

We have made corrections to the abstract, the introduction, and corrected remarks on the design.

Regional threshold values represent characteristics of variation series of actually measured hydrochemical parameters therefore, as a rule, do not exceed the accepted MAC and are though more strict, but flexible regulating values. At the same time, it should be noted that in some cases, for example, when assessing the content of substances of natural and anthropogenic origin, such as iron, manganese, and sulfate, such thresholds in some river basins may exceed MPC due to the specifics of formation of the natural local hydrochemical regime. As for the WHO standards, they are of recommendatory nature and are focused on drinking water, which does not reflect the ecological nature of regional thresholds.

Expert evaluation is a necessary element of the proposed system first of all due to the absence in it of automated spatial analysis of data distribution. The problem of spatial analysis will be solved during further development of the approach, in particular, by introducing the algorithm of estimation of geometrical distances between sampling points of estimated indicators and considering this variable during the clustering of different water bodies. Therefore, in the present study, homogeneity of clustering taking into account the spatial distribution of sampling points was evaluated by experts. In other words, the expert considers the quality of clustering by evaluating the clustering by clusters of the spatial localization of sampling locations represented by the data. In cases where the data in the cluster does not represent geographically contiguous localization, a decision is made about the lack of detail. Unfortunately, an effective IT solution for neural network generalization of such information as part of the developed system at this stage of research is not provided but is actively being developed. The current version of the system implements only automatic neural network estimation of temporal heterogeneity.

The second important function of expert estimation is the estimation of statistical homogeneity of data grouped into selected clusters. Such statistical evaluation was performed by determining the statistical significance of inter-cluster differences between the centers of distributions of each indicator. This stage has not yet been implemented in automatic mode and requires the participation of a specialist.

Due to the fact that these procedures are rather routine sequences of actions, a detailed description of the results of each step would significantly increase the volume and complicate the perception of the presented material, we have limited ourselves to a sufficient listing of necessary computational procedures, such as the Kruskal-Wallace and Mann-Whitney tests, whose separate description is beyond the essence and purpose of the study.

 This can be clearly shown by geoinformation methods, which, unfortunately, were not used in our work, as it was supposed to present in the article the basic scheme of using neural network modeling methods combined into a special analytical system. As it was already mentioned above, the spatial and physical-geographical differentiation of different types of waters was set by experts at the stage of detailed cluster analysis.  In the future, in the mentioned scheme of neural network modeling, external expert assessment will be completely replaced by an appropriate system algorithm.

Round 2

Reviewer 2 Report

There are still some aspects that should be improved by the authors that can be found below.

1. Introduction

- Line 167 -  extra spacing  General description of the idea (…) - Line 282, 283 and more – for example standardize mg/l for mg/L - Line 398, 399 and more – for example standardize mg / l for mg/L and mgO2 / l for mgO2/L

References

·       Standardize references (&). The authors have not checked all the references yet... see the case of reference 17

·       and other formatting, are outside the journal standards

Author Response

Introduction

Line 167 -  extra spacing  - Extra spaces removed.

General description of the idea (…) 

Line 282, 283 and more – for example standardize mg/l for mg/L 

Line 398, 399 and more – for example standardize mg / l for mg/L and mgO2 / l for mgO2/L

- All designations are standardized to the form mg/l, mgO2/l.

 References

- Standardize references (&). The authors have not checked all the references yet... see the case of reference 17

 Dear reviewer!

We ask for your help to understand the correct design of the references.

When designing, we used the "MDPI Reference List and Citations Style Guide" guide. According to this, we have issued links to articles in journals according to the rule:

Journal references must cite the full title of the paper, page range or article number, and digital object identifier (DOI) where available. Cited journals should be abbreviated according to ISO 4 rules, see the ISSN Center's List of Title Word Abbreviations or CAS's Core Journals List. Note: If you are not sure how to abbreviate a particular journal title, please leave the entire title. The Editorial Office will abbreviate those journal titles appropriately.

So, for example, link 17 is the article number, and link 38 is the page range. We have added DOI for publications with DOI, and links to sites if publications do not have DOI but are available online.

We did not find instructions on how to correctly refer to articles of laws, but we formatted it in accordance with the design of journal articles with a link to a page on the Internet.

We ask for your help to finally figure out the correct format of the design, if in the latest edition we again made a mistake in this.

Reviewer 3 Report

All the suggestions have been incorporated into the manuscript, which is now ready to be published.

Author Response

Dear reviewer!
We thank you for your attention to our article!